# The Effects of Biological Age on Speed-Explosive Properties in Young Tennis Players

**DOI:** 10.3390/jfmk8020048

**Published:** 2023-04-21

**Authors:** Filip Sinkovic, Dario Novak, Nikola Foretic, Erika Zemková

**Affiliations:** 1Faculty of Kinesiology, University of Zagreb, 10000 Zagreb, Croatia; filip.sinkovic@kif.unizg.hr (F.S.); dario.novak@kif.unizg.hr (D.N.); 2Faculty of Kinesiology, University of Split, 21000 Split, Croatia; nikolaforetic@hotmail.com; 3Department of Biological and Medical Sciences, Faculty of Physical Education and Sport, Comenius University in Bratislava, 814 69 Bratislava, Slovakia; 4Faculty of Health Sciences, University of Ss. Cyril and Methodius in Trnava, 917 01 Trnava, Slovakia

**Keywords:** peak height velocity, maturity offset, maturation, pre-puberty, puberty

## Abstract

Biological maturity can affect performance on motor tests, thus young players can have advantages or disadvantages during testing by being more or less mature than their peers of the same chronological age. The aim of this study was to investigate the effects of biological age on speed, agility, and explosive power in young tennis players. Fifty tennis players (age 12.3 ± 1.2 years, height 156.7 ± 12.8 cm, body mass 45.9 ± 8.9 kg) who were ranked up to 50th place in the ranking of the National Tennis Association, as well as up to 300th place on the international “Tennis Europe” ranking, participated in the study. They were divided into three groups according to their maturation status, that is, the peak height velocity (PHV) maturity offset: pre-PHV [<0] (*n* = 10); circa-PHV [0 to 1] (*n* = 10); and post-PHV [>1.5] (*n* = 25). They performed tests of speed (5, 10, and 20 m sprints), agility (20 yards, 4 × 10 yards, T-test, TENCODS, and TENRAG), and explosive power (countermovement jump, one-leg countermovement jump, squat jump, long jump, and one-leg triple jump). Results showed significantly higher height of the vertical jump in the post-PHV group compared to the pre-PHV group, in the range of ~16% to ~27%. Moreover, linear and change of direction speed was significantly faster in the post-PHV group compared to the pre-PHV group, in the range of ~5% to ~8%. Height of the squat jump and speed in the T-test were significantly better in the post-PHV group compared to the circa-PHV group, in the range of ~7% to ~15%, while height of the single-leg triple jump was significantly higher in the circa-PHV group compared to the pre-PHV group by ~7%. This study showed that tennis players of older biological age achieve better results in almost all variables of speed, agility, and explosive power compared to players of younger biological age. Coaches should be aware of the differences found in physical performance and consider the practical implications that maturation can have in the long-term development of young tennis players.

## 1. Introduction

The analysis of the tennis game points to the fact that tennis as a sport belongs to a group of technically more complex sports that require distinct technical, tactical, fitness, and psychological preparation from the players [1]. It follows from the definition itself that there are many movement structures and situations in tennis that indicate that the success of a tennis player is determined by the level and structure of a large number of abilities, knowledge, and traits [2]. Athletes, as well as their coaches, try every day to find new ways to improve their performance and results in individual sports like tennis. Taking into account the reactive demands of the game, the total duration of the match, the surface on which the game is played, and energy consumption, conditioning training of tennis players must be directed towards the development and maintenance of speed-explosive properties [3].

Speed, agility, and explosive power are very important for success in tennis. These abilities are treated together, due to several common characteristics. These include the use of the same energy resources, similar stimulation of the nervous system, and the need to meet the same preconditions for intensive training of an individual motor ability [4]. Athletes with more pronounced speed-explosive properties also more easily control their bodies in intensive training and competition situations, which greatly contributes to the game, as well as to the prevention of injuries [5]. A high level of these abilities is achieved through long-term preparation, proper planning, and programming of the training process through each phase of the athlete’s development.

More and more often, we come across the fact that conditioning training is effective even for children in pre-puberty [6]. Pre-puberty is a time of early anatomical adaptation of the heart, lungs, joints, and muscles to extended physical activities [6]. It is the foundation upon which athletes build demanding aerobic and anaerobic endurance for periods of specialization and peak performance [6]. The most favorable sensitive stages for the development of speed-explosive properties are the years immediately before puberty and +the years in the early phase of accelerated growth and development [6]. Since the quality of muscle and connective tissue is one of the basic conditions for efficient and fast movement, the period of pre-puberty and early puberty is the most favorable for intensive development of these abilities. In order to be able to carry out intensive conditioning training for improvement of these abilities in the early phase of accelerated growth and development, it is necessary to meet the conditions related to the adoption of movement techniques, the development of muscle and connective tissue, and the proprioceptive system [6]. A large number of competitions and increased training load represent a form of bodily stress for tennis players. This stress, in addition to improperly planned recovery, can lead to harmful consequences, i.e., injuries to their bodies. Lateral epicondylitis (tennis elbow) is the most frequent type of myotendinosis and can be responsible for substantial pain and loss of function of the affected limb [7]. Tennis biomechanics, player characteristics, and equipment are important in preventing the condition. Therefore, one of the most important factors in reducing the number of injuries is to optimally schedule load. This has become an indispensable part of planning and programming the training process, especially in the phases of pre-puberty and early puberty.

Biological age is the degree of biological aging of an organism and is estimated based on morphological, physiological, and functional indicators by comparison with their reference values [8]. Biological age does not have to be in accordance with the chronological years, which is often the case with children and adolescents. Biological age is estimated as the age at peak height velocity. This is the period in which an adolescent experiences the fastest growth in height, i.e., the time in which he/she grows the fastest during adolescent growth [9]. This determines the current level of physical development, that is, the one that is yet to follow, regardless of the child’s chronological age [10].

The main problem arises when it comes to biological age and sports in children and adolescents, as biological maturity affects performance on motor tests [11]. Children can have advantages or disadvantages during the tests by being more or less mature than peers of the same chronological age [11]. It is apparent who has extremely fast development of morphological features and functional abilities and who, in terms of the level of development, lags behind the norms of their age [11]. Biological age may affect motor abilities. For instance, handball players of older biological age achieve better results in speed and explosive power tests [11]. Similarly, better results in the change of direction speed tests are achieved by young hockey players [8], soccer players [11,12,13], futsal players [14], and tennis players [15] with older rather than younger biological age.

Considering the above results and the lack of research dealing with this issue in tennis, the aim of this paper is to investigate the effects of biological age on the performance of speed-explosive properties in young tennis players in the phases of pre-puberty and early puberty. It is expected that participants of older biological age achieve better results in almost all variables of speed, agility, and explosive power compared to those of younger biological age.

## 2. Materials and Methods

### 2.1. Participants

The sample included 50 young tennis players (with means of 12.3 ± 1.2 years, height 156.7 ± 12.8 cm, and weight 45.9 ± 8.9 kg) who are ranked up to 50th place in the National Tennis Association ranking, as well as up to 300th place on the international “Tennis Europe” ranking. The G-Power program (version 3.1.9.2; Heinrich Heine University, Dusseldorf, Germany) was used to estimate the appropriate number of participants with the expected effect power f = 0.33 m, alpha level of 0.05, and statistical power of 0.90. In addition, the sample size was confirmed and deemed satisfactory in relation to the actual study population and to the effect size of individual participants. The participants were divided into three groups according to their maturation status, that is, the peak height velocity (PHV) maturity offset: pre-PHV [<0] (*n* = 10); circa-PHV [0 to 1] (*n* = 10); and post-PHV [>1.5] (*n* = 25). Thus, a total of 45 respondents were included. Inclusion criteria included being in good health and physically active players who train at least three times per week and compete in regional, national, or international tournaments. Exclusion criteria were participants with PHV maturation offset [1 to 1.5] (*n* = 5) and any injury that influences tennis play and physical performance. The research was conducted in accordance with the Declaration of Helsinki and approved by the Ethics Committee of the Faculty of Kinesiology, University of Zagreb (protocol number 34; approval date 13 December 2021). All participants were familiar with the protocol and aim of the research, and the participants and their parents gave their written consent to participate. The complete testing protocol was explained to them in detail, with special emphasis on the fact that research requires a certain additional effort and presents a risk of injury that is the same as during a standard training process or competition.

### 2.2. Measurements

The biological age of the participants was assessed with body height (cm), sitting height (cm), body mass (kg), leg length (cm), and chronological age (years). The data obtained were entered into a specific regression equation for boys to determine PHV maturity offset: −9.236 + (0.0002708 × leg length × sitting height) + (−0.001663 × chronological age × leg length) + (0.007216 × chronological age × sitting height) + (0.02292 × ratio of body mass to body height) [10]. Therefore, a maturity offset of −1.0 indicates that the player was measured 1 year before peak height velocity, a maturity offset of 0 indicates that the player was measured at the time of peak height velocity, and a maturity offset of +1.0 indicates that the athlete was measured 1 year after peak height velocity. In accordance with that, age at peak high velocity (APHV) was calculated from an estimation between peak height velocity maturity offset and chronological age. The chronological age of the participants (years) was calculated by subtracting the date of birth from the date of measurement. Standing body height (cm) and sitting height (cm) were measured using a portable altimeter (Seca 213; seca gmbh, Hamburg, Germany). Leg length (cm) was calculated by subtracting the sitting height (cm) from the standing height (cm). Body mass (kg) was measured using a portable digital scale (Seca V/700; seca gmbh, Hamburg, Germany), while body fat percentage (%) was measured using the MALTRON BF 900 analyser (Maltron International Ltd., Rayleigh, UK).

Participants performed tests assessing speed (5, 10, and 20 m sprints), agility (20 yards, 4 × 10 yards, T-test, TENCODS, and TENRAG), and explosive power (countermovement jump (CMJ), single-leg countermovement jump (CMJ_L,R), squat jump (SJ), long jump (L_JUMP), and single-leg triple jump (SLTJ_L,R). Speed was measured with the Powertimer system (Newtest Oy, Oulu, Finland), agility with the SportReact system (SportReact, Zagreb, Croatia), and explosive power during jumps with the Optojump system (Microgate, Bolzano, Italy). Each test was performed three times, and the mean value of three trials was taken for further processing.

### 2.3. Study Design and Procedure

Prior to conducting the testing, all participants performed a standard tennis warm-up for a period of 15 min. The warm-up protocol included light-intensity running over 10 lengths of 20 m, followed by dynamic stretching exercises for a total of 15 min (lateral movements, skipping, jumping, lunges, and, finally, 4 lengths of sub-maximum acceleration). The warm-up was followed by tests of speed (5, 10, and 20 m sprints), agility (20 yards, 4 × 10 yards, T-test, TENCODS, and TENRAG), and explosive power (countermovement jump, one-leg countermovement jump, squat jump, long jump, and one-leg triple jump).

#### 2.3.1. Linear Sprint Speed Tests

Three electronic timing gates were positioned 5, 10, and 20 m from a predetermined starting line for the linear sprint speed tests. The subjects were instructed to begin with their preferred foot forward, placed on a line marked on the floor, and to run as quickly as possible along the 20-m distance from a stationary standing start. The times were recorded at 5 m (the first electronic timing gate), 10 m (the second electronic timing gate), and 20 m (the third electronic timing gate). For all tests, subjects performed three trials with 3–4 min of pause between the trials, and the mean value of three trials was taken for further processing. 

#### 2.3.2. Explosive Power Tests

In the countermovement and single-leg countermovement jump tests, hands were held at the hips to minimize the influence of the upper body on jump performance. From a standing position with straight knees, participants squatted down to ~90° and accelerated at maximal velocity in a vertical direction with both legs or a single leg. For all tests, subjects performed three trials with 1 min of pause between the trials, and the mean value of the three trials was taken for further processing. 

For the squat jump test, the starting position was with a knee flexion angle of 90°, torso straight, hands on hips, and feet shoulder-width apart. This position was maintained for 2 s before jumping. The push-off phase was executed while avoiding any kind of countermovement. During the apex of the jump phase, participants kept their legs fully extended. The landing phase occurred with both feet together in an upright position, with knees fully extended. For all tests, subjects performed three trials with 1 min of pause between the trials, and the mean value of the three trials was taken for further processing.

For the long jump test, all participants were instructed to perform a long jump from a standing position. Standardized instructions were given to participants that permitted them to begin the jump with bent knees and swing their arms to assist in the jump. A line drawn on a hard surface served as the starting line. The length of the jump was determined using a tape measure, which was affixed to the floor. For all tests, subjects performed three trials with 1 min of pause between the trials, and the mean value of the three trials was taken for further processing.

In the single-leg triple jump test, the participants began by standing on the designated test leg, with their toe on the starting line. When ready, the participants performed three consecutive maximal jumps forward with the designated leg. Upper extremity movement during single-leg horizontal hop testing was not restricted, although the participants were instructed to “stick” the landing on the last jump. After the practice trials, three test trials were performed on each leg in alternating order. A 30-s rest period was allowed between practice and test trials. The mean distance of the three test trials for each leg was calculated.

#### 2.3.3. Change of Direction Speed and Reactive Agility Tests

In the 20 yards test, the examinee started in a three-point stance and ran 5 yards in one direction, ran 10 yards in the opposite direction, and then sprinted back to the starting point. This exercise tests lateral speed and coordination. The timing began on a sound signal and stopped when the subject passed through the timing gate on their return. The time was measured in hundreds of seconds.

In the 4 × 10 yards test, parallel lines were drawn on the floor 10 yards apart. The subjects had to run back and forth four times as fast as possible between the starting line and the other line, crossing each line with both feet every time. The timing began on a sound signal and stopped when the subject passed through the timing gate on their return. The time was measured in hundreds of seconds.

For the T-test, four cones were arranged in a T shape, with one cone placed 9.14 m from the starting cone and two additional cones placed 4.57 m from either side of the second cone. All of the times were recorded using an electronic timing gate with a height of 0.75 m and a width of 3 m, in line with the marked starting point. The subjects were asked to sprint forward 9.14 m from the start line to the first cone and touch the tip with their right hand, shuffle 4.57 m left to the second cone and touch the tip with their left hand, shuffle 9.14 m to the right to the third cone and touch the tip with their right hand, and shuffle 4.57 m back left to the middle cone and touch the tip with their left hand, before finally backpedaling to the start line. The timing began on a sound signal and stopped when the subject passed through the timing gate on their return. The trials were deemed unsuccessful if the participants failed to touch a designated cone, crossed their legs while shuffling, or failed to face forward at all times. The time was measured in hundreds of seconds.

The sport-specific change of direction speed (TENCODS) and reactive agility (TENRAG) variables were measured using tests that exhibit very good metric characteristics and are reliable and valid [16]. The tests were measured using the SportReact system (SportReact, Zagreb, Croatia) made up of laser tape sensors and LED screens with different signs and colors (16). The pre-planned ability to change direction (TENCODS) and reactive agility (TENRAG) tests are constructed in such a way that the participants imitate specific movements in tennis (Figure 1). In both tests, participants start from a predetermined starting line. When the infrared signal (IR1) located next to the starting line is interrupted by the “split step,” the time starts to be measured, and one of the two lights (L1 or L2) lights up. The participant should recognize which light has turned on, run with overstepping and a lateral side to side technique to the stand with a ball placed on it (S1 or S2), and hit the aforementioned ball forehand or backhand in front of the body with enough force that the ball hits the ground. After playing the shot, the player should return as quickly as possible to the device in front of the starting line and interrupt the infrared signal (IR2) again, which ends the measurement. In the pre-planned change of direction speed test (TENCODS), the subjects know in advance which light will turn on; that is, they can plan in advance to run and play forehand or backhand shots. Each test was performed nine times with a 60-s intermission time between the measurement particles, and the mean measured value for both tests was taken for further processing [16].

### 2.4. Statistical Analysis

The obtained data were processed in the program Statistica 14.0.1.25 (TIBCO software Inc, CA, USA) for the Windows operating system and in Microsoft Excel 2016 (Palo Alto, CA, USA). Basic descriptive parameters (mean—$\bar{x}$; standard deviation—SD) were used to describe each variable. The normality of the distribution was tested with the Shapiro-Wilk W test. One-way analysis of variance for independent samples (ANOVA) was used to investigate the effects of biological age on performance on motor ability tests, and the Bonferroni post-hoc test was used for a more detailed interpretation of the results. In addition, multivariate analysis of variance (MANOVA) was used for more precise conclusions. Effect size (Cohen’s d) was calculated to assess differences in the tested variables. Thresholds for effect size were statistically set to the following parameters: insignificant (<0.35), small (0.35–0.80), medium (0.80–1.5), and large (>1.5). The level of statistical significance was set at *p* < 0.05.

## 3. Results

Table 1 shows descriptive parameters that include chronological age (years), body height (cm), body mass (kg), body fat (%), age at peak height velocity (years), and peak height velocity maturity offset (years). The Shapiro-Wilk W test showed a normal distribution, which enabled further statistical processing.

Table 2 shows the performance results of speed-explosive tests with regard to the biological age of the subjects. Vertical jump height (CMJ, SJ, L_JUM, SLTJ_L, SLTJ_R) was significantly higher in older maturational groups compared to subjects of younger biological age. Also, linear sprint speed (SP10m, SP20m), change of direction speed (T-test), and reactive agility speed (TENRAG) were significantly faster in older maturational groups compared to subjects of younger biological age. There were no significant differences in the variables (CMJ_L, CMJ_R, SP5m, AG20Y, AG4x10Y, and TENCODS) with regard to groups of different biological ages. A more detailed analysis revealed that the variables (CMJ, SJ, L_JUM, SLTJ_L, SLTJ_R, SP10m, SP20m, and T-test) differ significantly in the pre-PHV group compared to the post-PHV group. Also, the variables (SJ and T-test) differ significantly in the pre-PHV group compared to the circa-PHV group, while the variable (SLTJ_L) differs significantly in the circa-PHV group compared to the post-PHV group.

Table 3 shows the multivariate analysis of variance (MANOVA) between maturational groups.

## 4. Discussion

A more detailed analysis of findings revealed that participants of older biological age achieved better results in almost all variables of speed, agility, and explosive power compared to those of younger biological age. Vertical jump height (countermovement jump ~21%, squat jump ~27%, long jump ~5%, one-leg triple jump ~16% to ~25%) was significantly higher in older maturational groups compared to subjects of younger biological age. Also, linear sprint speed (10 and 20 m sprints ~5% to ~8%), change of direction speed (T-test ~7%), and reactive agility speed (TENRAG ~7%) were significantly faster in older maturational groups compared to subjects of younger biological age. However, no significant differences were obtained in the variables (one-leg countermovement jump, 20 yards, 4 × 10 yards, and TENCODS) with regard to groups of different biological ages. These findings are consistent with the assumptions that biological maturity affects the performance of motor tasks and that participants can have advantages or disadvantages in tests by being more or less mature than peers of the same chronological age [11].

The results of the anthropometric tests indicate that participants of younger biological age have lower body height, muscle mass, and percentage of fat compared to those of older biological age. Such results were also confirmed in previous research on a large sample of 902 young tennis players (11–16 years old), where it was found that participants in the pre-PHV group have a lower body height (~7% to 12%) and a lower body mass (~18% to ~30%) from the circa- and post-PHV groups [17]. These differences in body structure could be one of the main factors that lead to differences in motor test results. In this research, height of the vertical jump tests (countermovement jump, squat jump, long jump, and one-leg triple jump) was significantly higher in the post-PHV group compared to the pre-PHV group, in the range from ~16% to ~27%. Moreover, speed in linear and change of direction tests (10, 20 m sprints and T-test) was significantly faster in the post-PHV group compared to the pre-PHV group, in the range from ~5% to ~8%. Height of the squat jump and speed in T-test were significantly better in the post-PHV group compared to the circa-PHV group, in the range from ~7% to ~15%, while height of the one-leg triple jump was significantly higher in the circa-PHV group compared to the pre-PHV group, by ~7%.

Similar results were obtained in previous research, where, in a sample of 112 young football players of pre-pubescent age, better results on tests of motor abilities (~11–14%) were achieved in the post-PHV group [18]. Similar research was also conducted on a sample of 155 young tennis players with an average age of 13.1 ± 2.2 years, where lower results on tests of explosive power (~15–20%) and speed (~7–12%) were achieved in the pre- and circa-PHV groups compared to the post-PHV group [11,19].

As preplanned change of direction speed and reactive agility are different and independent abilities that are influenced by many factors such as observation, perception, anticipation, and decision-making [20,21], this is one of the first researches in tennis that has led to new insights by indicating how subjects in the pre-PHV group achieve lower results in preplanned and reactive agility tests compared to the circa- and post-PHV groups. 

Limitations of this study are discussed below. First, the subjects involved in this study were youth tennis players in a very sensitive and crucial developmental phase. In addition, the motor tests were conducted with a convenient sample of subjects under controlled conditions, and the results might have been different if the tests had been conducted on a different tennis surface. Moreover, we evaluated a larger sample population of post-PHV players than pre-PHV players, which may have influenced the statistical significance of the established relationships. Therefore, detailed effects of biological age on speed-explosive properties in young tennis players should be more precisely studied through longitudinal investigations. However, we believe that the present design may offer a starting point to suggest practical applications to tennis professionals. Our findings provide useful information for coaches to create a wide range of tennis-specific situations to develop proper performance, especially for their players’ neuromuscular fitness. Also, this research has shown the simplicity of monitoring, collecting, and analyzing data, which can be applied and used in regular tennis training. The future research should conduct tests with subjects of different genders and different competition categories in order to obtain the best and most accurate data that would enable even greater practical and scientific applications. Also, future research needs to include larger samples with higher performance levels, which could help prevent potential selection bias.

## 5. Conclusions

The results of this research support the assumption that tennis players of older biological age achieve better results in almost all variables of speed, agility, and explosive power compared to those of younger biological age. This study also showed that biological maturity affects performance on motor tests and that children can have advantages or disadvantages during testing by being more or less mature than their peers of the same chronological age. Therefore, coaches should be aware of the differences found in physical performance and consider the practical implications that maturation can have on the long-term development of young tennis players. Recommendation for additional studies is directed to determine the best training approaches and content (specific to each maturity stage) to meaningfully improve neuromuscular performance in young tennis players in the phases of pre-puberty and early puberty.

## Figures and Tables

**Figure 1 jfmk-08-00048-f001:**
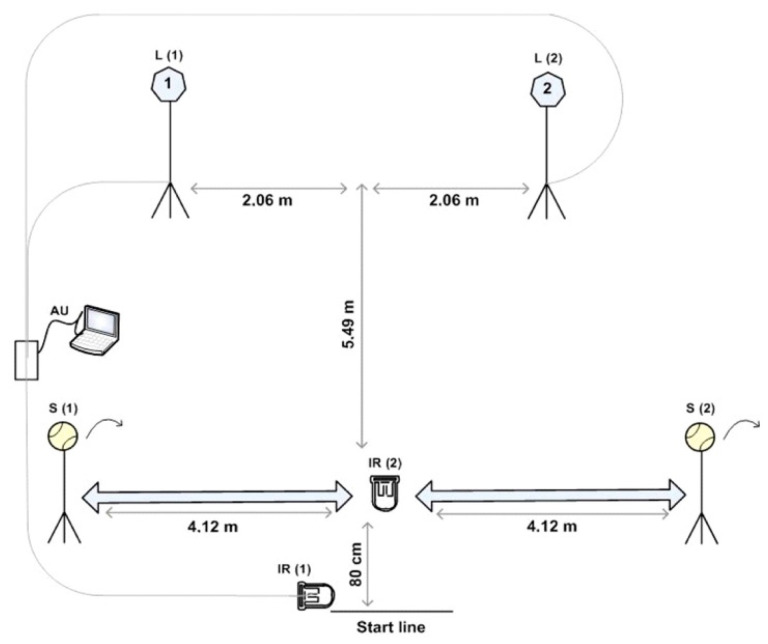
TENCODS and TENRAG tests.

**Table 1 jfmk-08-00048-t001:** Descriptive parameters according to players’ maturational groups.

Variables		Maturational Groups		
	Players (*n* = 45)	Pre-PHV (*n* = 10)	Circa-PHV (*n* = 10)	Post-PHV (*n* = 25)
Chronological age (years)	12.1 ± 1.3	11.2 ± 0.9 ^### $$^	12.8 ± 0.7	13.6 ± 0.3
Body height (cm)	157.6 ± 10.4	150.4 ± 5.2 ^### $$^	161.7 ± 3.0 ^#^	171.7 ± 8.0
Body mass (kg)	45.8 ± 9.1	39.9 ± 4.7 ^### $$^	48.1 ± 5.6 ^##^	58.2 ± 6.5
Body fat (%)	15.5 ± 4.6	15.3 ± 4.7	15.6 ± 5.5	15.7 ± 3.6
APHV (years)	13.4 ± 0.6	13.4 ± 0.6	13.5 ± 0.5	13.1 ± 0.5
Maturity offset (years)	1.3 ± 1.2	2.2 ± 0.5 ^### $$^	0.77 ± 0.1 ^###^	−0.4 ± 0.4

Legend: Pre-PHV—pre peak height velocity group; Circa-PHV—around peak height velocity group; Post-PHV—post peak height velocity group; APHV (years)—estimated age at peak height velocity; ^$$^ significantly different from Circa-PHV group (*p* < 0.001); ^#^ significantly different from Post-PHV group (*p* < 0.05); ^##^ significantly different from Post-PHV group (*p* < 0.01); ^###^ significantly different from Post-PHV group (*p* < 0.001).

**Table 2 jfmk-08-00048-t002:** Differences between maturational groups in vertical jump height, linear sprint speed, change of direction speed, and reactive agility.

	Maturational Groups			One-Way ANOVA	Effect Size (90% CI)		
Variables	Pre-PHV (*n* = 10)	Circa-PHV (*n* = 10)	Post-PHV (*n* = 25)	*p*-Value	Pre-PHV vs. Circa-PHV	Circa-PHV vs. Post-PHV	Pre-PHV vs. Post-PHV
**Vertical jump height**							
					0.79	0.80	1.60
CMJ (cm)	20.1 ± 2.6 ^###^	22.3 ± 2.9	24.7 ± 3.0	<0.001 *	(0.21–1.37)	(0.03–1.56)	(0.96–2.24)
					0.14	0.86	0.90
CMJ_L (cm)	10.3 ± 1.6	10.5 ± 1.2	11.6 ± 1.3	<0.08	(−0.4–0.7)	(0.09–1.63)	(0.31–1.48)
					0.51	0.45	0.97
CMJ_R (cm)	10.2 ± 1.5	11.0 ± 1.4	11.6 ± 1.2	<0.05 *	(−0.04–1.08)	(−0.29–1.19)	(0.38–1.56)
					0.63	1.10	1.99
SJ (cm)	19.6 ± 2.3 ^$ ###^	21.4 ± 3.4	25.1 ± 3.2	<0.001 *	(0.06–1.2)	(0.31–1.89)	(1.31–2.67)
					0.12	0.89	1.03
L_JUM (cm)	149.1 ± 14.6 ^##^	150.9 ± 16.4	167.2 ± 20.0	<0.02 *	(−0.43–0.67)	(0.12–1.66)	(0.44–1.62)
					0.53	1.18	1.78
SLTJ_L (cm)	402.2 ± 47.5 ^###^	430.6 ± 58.0	503.1 ± 64.6 $	<0.001 *	(−0.02–1.10)	(0.38–1.97)	(1.12–2.43)
					0.55	1,09	0.61
SLTJ_R (cm)	406.8 ± 44.5 ^##^	433.0 ± 51.9	470.7 ± 69.5	<0.01 *	(−0.01–1.12)	(0.49–1.69)	(−0.13–1.36)
**Linear sprint speed**							
					−0.54	0.04	−0.9
SP5m (s)	1.3 ± 0.1	1.2 ± 0.1	1.2 ± 0.1	<0.06	(−1.10–0.02)	(−0.34–1.14)	(−1.48–0.32)
					−0.74	−0.73	−1.50
SP10m (s)	2.1 ± 0.1 ^###^	2.1 ± 0.1	2.0 ± 0.1	<0.001 *	(−1.31–0.16)	(−1.49–0.02)	(−2.13–0.87)
					−0.64	−1.39	−0.86
SP20m (s)	3.7 ± 0.2 ^###^	3.7 ± 0.1	3.5 ± 0.2	<0.001 *	(−1.21–0.07)	(−2.01–0.77)	(−1.63–0.09)
**Change of direction speed**							
					0.29	−0.99	−0.63
AG20Y (s)	5.5 ± 0.3	5.6 ± 0.2	5.3 ± 0.3	<0.11	(−0.26–0.85)	(−1.77–−0.21)	(−1.20–−0.06)
					−0.20	−0.45	−0.72
AG4x10Y (s)	10.5 ± 0.4	10.4 ± 0.6	10.1 ± 0.6	<0.17	(−0.76–0.35)	(−1.2–0.29)	(−1.3–0.15)
					0.11	−1.43	−1.23
T-TEST (s)	12.1 ± 0.5 ^$$ ###^	12.0 ± 0.4	11.3 ± 0.8	<0.001 *	(−0.44–0.66)	(−2.26–−0.61)	(−1.83–−0.62)
					−0.28	−0.54	−0.77
TENCODS (s)	3.3 ± 0.2	3.2 ± 0.1	3.1 ± 0.2	<0.13	(−0.84–0.27)	(−1.29–0.20)	(−1.35–−0.19)
**Reactive agility speed**							
					0	0.94	0.96
TENRAG (s)	3.2 ± 0.2	3.2 ± 0.2	3.0 ± 0.2	<0.04 *	(−0.55–0.55)	(0.16–1.71)	(0.38–1.55)

Legend: Pre-PHV—pre peak height velocity group; Circa-PHV—around peak height velocity group; Post-PHV—post peak height velocity group; CMJ (cm)—countermovement jump with arms set on hips; CMJ_L (cm)—single leg (left) countermovement jump with arms set on hips; CMJ_R (cm)—single leg (right) countermovement jump with arms set on hips; SJ (cm)—squat jump; L_JUM (cm)—long jump; SLTJ_L (cm)—single leg (left) triple jump; SLTJ_R (cm)—single leg (right) triple jump; SP5m (s)—5m linear sprint; SP10m (s)—10m linear sprint; SP20m (s)—20m linear sprint; AG20Y (s)—20 yards agility test; AG4x10Y (s)—4 × 10 yards agility test; T-TEST (s)—agility T-TEST; TENCODS (s)—change of direction speed test; TENRAG (s)—reactive agility test; * significant interaction (*p* < 0.05); ^$^ significantly different from Circa-PHV group (*p* < 0.05); ^$$^ significantly different from Circa-PHV group (*p* < 0.001); ^##^ significantly different from Post-PHV group (*p* < 0.01); ^###^ significantly different from Post-PHV group (*p* < 0.001).

**Table 3 jfmk-08-00048-t003:** Multivariate analysis of variance (MANOVA) between maturational groups.

Variables	F	*p*
Vertical jump height variables	2.78	0.002 *
Linear sprint speed variables	2.40	0.04 *
Change of direction speed variables	3.48	0.001 *
Specific agility speed variables	2.00	0.11

Legend: *— statistical significance (*p* < 0.05).

## Data Availability

Data available on request.

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
