# Peer review of "The Effects of Biological Age on Speed-Explosive Properties in Young Tennis Players"

_jfmk, 2023, doi:10.3390/jfmk8020048_

Round 1

Reviewer 1 Report

Dear Authors,

First of all, I would like to congratulate you for your efforts in carrying out this very interesting research. It is a novel object of study, of interest and of clinical impact for athletes and professionals who relate to them.

However, there are formal and methodological errors in this manuscript that should be addressed before its possible publication in this Journal.

ABSTRACT:
There is an abuse of abbreviations. The authors should eliminate them.
It should begin with a sentence on the state of the art.
The section should end with the clinical or practical impact of the results obtained.

INTRODUCTION:
Perhaps an approach to the most frequent problems among players of this sport should be addressed (doi: https://pubmed.ncbi.nlm.nih.gov/34397403).

METHODS:
The applied study design should be made explicit.
The sample size should be calculated in relation to the real study population and in relation to the effect size of the specific participants.
There are grammatical errors and typos that should be corrected throughout the entire manuscript since this division into subsections in section 2.3. is not at all clear (page 3, line 133).

RESULTS:
The p value is asymptotic, please correct "0.00" to "0.001" or whatever figure specifically corresponds.
The bivariate statistical analysis should be complemented with some other multivariate techniques to draw more solid conclusions.
In addition, the tests applied (as well as possible new tests to be applied) should be complemented with the size of their effects.

DISCUSSION:
There is an abuse of abbreviations and repetition of information already conveyed in Results.
The methodological limitations of this research should be acknowledged more honestly and openly.
The practical implications of the results obtained should be more thoroughly described.

Kind regards

Author Response

Sincerely,

Filip Sinković

Reviewer 2 Report

This study may provide valuable information for sports specialists and tennis coaches who want to optimize the training of their players. The study focuses on the influence of biological age on speed, agility, and explosive strength in young tennis players. Various well-chosen tests were used to evaluate these parameters, and the players were divided into three groups according to their degree of biological maturity. But the method does not specify how the PHV is determined. And how was the difference between age and PHV calculated - by biological age or calendar age, it is that not mentioned? This is a very important clarification, because all the results and conclusions in the article are based on the division of groups.

table 1 - the method lacks information on the method of determining the percentage of body fat, how the APHV is calculated

Author Response

Sincerely,

Filip Sinković

Round 2

Reviewer 1 Report

Dear Authors,

The research presented is well conducted and interesting in relation to the construct it studies and the sample it analyzes. Furthermore, after the changes and improvements made by the authors, I believe that the manuscript should be accepted for future publication.

Kind regards